# The Changes in the Quantity of Lymphocyte Subpopulations during the Process of Sepsis

**DOI:** 10.3390/ijms25031902

**Published:** 2024-02-05

**Authors:** Jiale Yang, Xiaojian Zhu, Jun Feng

**Affiliations:** 1Department of Emergency Medicine, Tongji Hospital, Tongji Medical College, Huazhong University of Science and Technology, Wuhan 430074, China; m202376587@hust.edu.cn; 2Department of Critical Care Medicine, Tongji Hospital, Tongji Medical College, Huazhong University of Science and Technology, Wuhan 430074, China; 3Department of Hematology, Tongji Hospital, Tongji Medical College, Huazhong University of Science and Technology, Wuhan 430074, China; zhuxiaojian@hust.edu.cn

**Keywords:** sepsis, lymphocyte depletion, revive, absolute lymphocyte count, innate lymphocyte subgroups, effector T lymphocytes, B lymphocytes, dendritic cells

## Abstract

Sepsis remains a global challenge, especially in low- and middle-income countries, where there is an urgent need for easily accessible and cost-effective biomarkers to predict the occurrence and prognosis of sepsis. Lymphocyte counts are easy to measure clinically, and a large body of animal and clinical research has shown that lymphocyte counts are closely related to the incidence and prognosis of sepsis. This review extensively collected experimental articles related to lymphocyte counts since the unification of the definition of sepsis. The article categorizes and discusses the relationship between absolute lymphocyte counts, intrinsic lymphocyte subsets, effector T-lymphocytes, B-lymphocytes, dendritic cells, and the incidence and prognosis of sepsis. The results indicate that comparisons of absolute lymphocyte counts alone are meaningless. However, in addition to absolute lymphocyte counts, innate lymphocyte subsets, effector T-cells, B-lymphocytes, and dendritic cells have shown certain research value in related studies.

## 1. Introduction

Sepsis is a common disease in the ICU and can easily lead to poor outcomes in patients due to multiple organ failure. In 2017, member states of the World Health Organization (WHO) declared improving the prevention, recognition, and treatment of sepsis a global health priority [1]. The latest epidemiological statistics show a decrease in the incidence rate of sepsis in recent years, but there is still a lack of effective data in low- and middle-income countries [2]. Another study from the same year shows that the incidence of sepsis has not declined in recent years [3]. This suggests that the incidence rate of sepsis may be underestimated while also indicating a lack of effective means for the early recognition and diagnosis of sepsis in low- and middle-income countries. In this scenario, there is an urgent need for highly adaptable biomarkers to identify sepsis, assess prognosis, and guide treatment plans. As our understanding of the pathophysiology of sepsis has deepened, there have been changes in the definition and treatment strategies of sepsis among clinical physicians. Initially, Bone’s definition emphasized sepsis as a systemic response to infection, characterized by an exaggerated inflammatory response. Clinical practice has found that during infection, patients’ blood contains many cytokines [4]. Subsequently, Clark and others [5] proposed the hypothesis that these cytokines may be the cause of tissue damage and symptoms such as shock observed during sepsis. This phenomenon is referred to as cytokine storm and has garnered significant attention from clinical physicians. Unfortunately, up until now, all clinical trials aimed at suppressing the inflammatory response or targeting cytokines have failed [6]. Faced with this disappointing reality and a further understanding of the pathophysiology of sepsis, many have turned their attention towards new directions. The new definition of sepsis (sepsis 3.0) introduced in 2016 highlights sepsis as a life-threatening organ dysfunction caused by a dysregulated host response to infection [7]. Clinical practice has found that patients who survive early sepsis often experience nosocomial infections, with most of the infections being caused by opportunistic pathogens. Additionally, latent viruses may be reactivated [8,9]. Based on this, clinicians speculate that after the early hyperinflammatory state, the body enters a state of low inflammation accompanied by persistent immune suppression [10]. This phenomenon is known as Compensatory Anti-Inflammatory Response Syndrome (CARS) (Figure 1) [11]. However, recent studies have shown that during the early stages of sepsis, both pro-inflammatory and anti-inflammatory responses occur simultaneously [12].

Existing meta-analysis has shown that the neutrophil-to-lymphocyte ratio may be a useful prognostic biomarker for patients with sepsis [13]. To further understand the changes and the role of the immune system during sepsis, we focus on lymphocytes, and we have reviewed the related research on sepsis lymphocytes since the establishment of a unified definition of sepsis. The review aims to explore the alterations in lymphocytes during sepsis and their significance.

## 2. Lymphocytes and Their Subgroups

The immune system of the body is divided into innate immunity and adaptive immunity. Innate immunity primarily serves as the first line of defense through direct killing by phagocytic cells. Among them, innate lymphocyte subgroups (NK cells, γ-δ T cells, and CD4+CD25+ T cells) play an upstream regulatory role in the functions of macrophages, dendritic cells, and T lymphocytes [14]. Adaptive immunity targets specific antigens and involves cell-mediated immunity mediated by effector T lymphocytes (CD4+ T lymphocytes and CD8+ T lymphocytes) and humoral immunity mediated by B lymphocytes. It serves as the main component of host immune responses. In addition, dendritic cells (DCs) serve as crucial bridge cells between innate and adaptive immunity as antigen-presenting cells.

This article is elaborated in the following order: (1) absolute lymphocyte count, (2) innate lymphocyte subgroups, (3) effector T lymphocytes, (4) B lymphocytes, and (5) dendritic cells.

### 2.1. Absolute Lymphocyte Count

A total of 8 relevant articles were included [15,16,17,18,19,20,21,22]. The reference numbers and related information for these nine articles are listed in Table 1. Based on the common findings from the 8 articles, the following conclusions can be drawn: (a) the absolute lymphocyte count in septic patients is significantly lower compared to healthy adults; (b) the absolute lymphocyte count in septic patients who died is significantly lower compared to those who survived. These conclusions demonstrate the correlation between lymphocyte count and the mortality of sepsis, providing a potential indicator for predicting sepsis and its prognosis through absolute lymphocyte count. Furthermore, Francois et al. [23] conducted experiments that demonstrated the ability of human IL7 (CYT107) to reverse lymphocyte depletion in septic shock patients, resulting in a 3-to 4-fold increase in absolute lymphocyte count as well as circulating CD4+ and CD8+ T cell counts.

### 2.2. Congenital Lymphocyte Subsets

Congenital lymphocyte subsets include natural killer cells (NK cells), regulatory T cells, and γδ T cells (gamma delta T cells). These lymphocyte subsets are present during the early stages of human development. Unlike other lymphocyte subsets in the acquired immune system, their ability to combat pathogens is innate. They play a crucial role in the immune function of infants and other populations.

#### 2.2.1. Gamma-Delta T Cells (γ-δT Cells)

γ-δT cells are members of the T lymphocyte subset [24], but due to their critical role in innate immunity [25], this review includes them in the description of congenital lymphocyte subsets. In early animal experiments, researchers observed that peripheral blood γ-δT cells can recognize heat shock proteins to initiate innate immunity after injury [26]. Meanwhile, γ-δT cells in mucosal tissues attract T lymphocytes by releasing chemotactic factors [27]. A subsequent retrospective clinical study revealed a decrease in the number of γ-δT cells in the peripheral blood of patients with trauma and sepsis [28]. Similar results were observed in a prospective clinical trial [29]. The research findings suggest a protective role of γ-δT cells in the human body, but the specific mechanisms are still unclear. There are reports suggesting that γ-δT cells have antigen-presenting cell (APC) functions [30,31], which has become a focal point in studying the protective mechanisms of γ-δT cells. The latest study [32] by Yang, Xunwei, and colleagues assessed the antigen-presenting function of γ-δT cells in sepsis patients. Peripheral blood γ-δT cells from sepsis patients were collected for cell-based assays. The experimental results showed that the ability of sepsis patient γ-δT cells to receive amplification signals after antigen stimulation was impaired. Additionally, the expression of APC markers did not increase after stimulation, adhesive capacity significantly decreased, and the ability to induce the proliferation of T lymphocytes was lost. All of the experimental results indicate that the peripheral blood γ-δT cells in sepsis patients have impaired APC function. Treating this impairment is theoretically feasible, but progress has been slow [33]. It is only recently that some achievements in immunotherapy based on γ-δT cells have emerged [34,35,36]. Prior to this, Yuan, Fawei, and colleagues [37] proposed a different viewpoint, suggesting that γ-δ1T cells play an immune suppressive role in sepsis. γ-δT cells can be divided into two subsets: γ-δ1T cells and γ-δ2T cells [38,39]. These two subsets have distinct spatial distributions and physiological functions. Previous experiments have demonstrated that γ-δ1T cells play an immunosuppressive role in autoimmune diseases and tumors [37]. Yuan, Fawei, and colleagues speculate that a similar phenomenon may exist in sepsis. Experimental results indicate that the proportion of peripheral blood γ-δ1T cells in sepsis patients is lower compared to the healthy control group. Furthermore, within the sepsis patient group, the proportion of peripheral blood γ-δ1T cells is even lower in shock patients compared to non-shock patients. Additionally, the experiment revealed that an increased proportion of peripheral blood γ-δ1T cells inhibits the proliferation of T lymphocytes and suppresses the secretion of IFN-γ by T lymphocytes, suggesting the immunosuppressive capability of γ-δ1T cells. The research conducted by Yuan, Fawei, and colleagues presents a new direction in the study of γ-δT cells, and we look forward to further developments in this field.

#### 2.2.2. Regulatory T Cells

Regulatory T cells (Treg cells) are a subset of T lymphocytes characterized by the expression of CD4+CD25+ on their surface. Due to their specialized regulatory role in the immune system, they are classified as innate lymphoid cells. Under physiological conditions, Treg cells maintain normal immune homeostasis by secreting immunosuppressive cytokines such as IL-10 and Transforming Growth Factor Beta 1 (TGF-β1) [40,41]. Therefore, it is speculated that Treg cells participate in the development of immunosuppression in sepsis, and relevant experiments have validated this hypothesis [42]. Peripheral blood samples were collected from sepsis patients after burns, and Treg cells were isolated using a CD4+CD25+ regulatory T cell isolation kit. Flow cytometry was then used to measure the purity of Treg cells expressing Foxp-3 through staining with an anti-human Foxp-3-FITC antibody. Finally, the supernatant was collected to measure the levels of IL-10 and TGF-β1. The results showed significantly higher overall expression of Foxp-3 in the severe burn, sepsis, and death groups. Consistently, the cytokines IL-10 and TGF-β1 also showed elevated levels. Foxp-3 is crucial for the differentiation and function of Treg cells [43]. The above results seem to support the notion that regulatory T cells play a negative regulatory role in the process of sepsis, which may be harmful to the body. However, subsequent animal experiments have challenged this viewpoint. Kühlhorn, Franziska, and others [44] used DEREG mice (depletion of regulatory T cells) to create a cecal ligation and puncture (CLP) model of sepsis. The results showed that Treg depletion worsened late-stage survival, suggesting a beneficial role of Foxp-3+ Treg cells in severe sepsis. By collecting more clinical trial data [45,46], we have discovered that in the early stages of sepsis, the proportion of Foxp-3+ Treg cells is the same between the survivor group and the non-survivor group, but the absolute number is higher in the former. On the other hand, in the later stage of sepsis (three days later), the survivor group exhibits an increased absolute count of Foxp-3+ Treg cells, while the proportion is comparatively lower.

#### 2.2.3. Natural Killer Cells (NK Cells)

NK cells are essential components of the innate immune system and are capable of killing target cells in their unactivated state [47]. Their role in combating infections and tumors has been widely recognized [48]. Through the study of changes in NK cells in collected cases of sepsis, we have observed two contradictory outcomes. One outcome supports the notion that NK cells are a risk factor for sepsis. Animal experiments conducted by Sherwood et al. [49] demonstrated that mice with depleted NK cells exhibited improved survival rates following cecal ligation and puncture. A prospective clinical trial conducted by David Andaluz-Ojeda et al. [50] showed that the absolute count and relative concentration of NK cells in survivors of sepsis were lower than those of deceased patients. According to the analysis of survival curves, it was observed that an NK cell count exceeding 83 cells/mm3 on the first day was linked to early mortality. De Pablo et al. [47] further supported these findings, indicating that patients with the highest NK cell counts in sepsis had the lowest probability of survival. However, another outcome supports the notion that NK cells are a protective factor in sepsis. Clinical trial results from Giamarellos-Bourboulis et al. [51] showed a significant increase in NK cells during the early stages of sepsis. Additionally, patients with NK cells accounting for ≥20% of the total lymphocyte count had longer survival times compared to patients with NK cells accounting for <20% of the total lymphocyte count. In addition to the previously mentioned clinical trials, studies conducted by Boomer et al. [49] and Holub et al. [50] also reported a substantial decrease in NK cells during the early stages of sepsis, specifically within 48 h. This decrease in NK cells may be associated with secondary nosocomial infections and adverse outcomes in patients.

### 2.3. Effector T Cells

Effector T cells are a subset of T lymphocytes composed of CD4+ T cells and CD8+ T cells that differentiate from αβ T cells. They are the main component of lymphocytes in the peripheral circulation [52]. 

According to the findings from the included clinical studies [15,22,49,51,53,54,55,56,57,58,59,60], the following common conclusions can be drawn: (a) Effector T cells show a significant decrease during the early stages of sepsis, even before clinical manifestations occur. (b) Compared to sepsis survivors, deceased patients have lower counts of effector T cells (mostly observed in CD4+ T cells). (c) The number of effector T cells tends to recover in the later stages of sepsis (around 7 days after sepsis diagnosis).

There is still controversy regarding the mechanisms of effector T lymphocyte recovery. In a comparison of lymphocyte subset changes in sepsis induced by Gram-positive and Gram-negative bacteria, Holub et al. [58] attributed the rapid recovery of T cell counts to the inhibition of bacterial growth and cessation of stress responses. This viewpoint aligns with the theory of cell migration, which suggests that the decrease in lymphocyte count during sepsis is due to the migration of many lymphocytes to the site of infection. This theory supports the occurrence of compensatory anti-inflammatory response syndrome (CARS) following excessive inflammation. On the other hand, Tomino et al. [56] proposed that lymphocyte dysfunction occurs early in sepsis. They categorized sepsis patients into four quadrants (ABCD) based on the percentage of PD-1 (programmed cell death protein 1) on T cells < 19.5% and TCR (T-cell receptor) count > 78.3%. By comparing the data on days 1, 3, and 7 after enrollment, the results are shown in Figure 2 [61]. It can be observed that in the three cases of early sepsis with high expression of PD-1 leading to death, PD-1 levels consistently exceeded 19.5%. This finding supports the theory of cell apoptosis, suggesting that lymphocyte reduction during sepsis is due to the excessive apoptosis of lymphocytes. This apoptosis is present from the early stages of sepsis, which contradicts the compensatory anti-inflammatory response syndrome theory. Therefore, this supports the claim that anti-inflammatory and pro-inflammatory responses occur simultaneously, without a sequential order. Different viewpoints will lead to different directions in treatment, and we look forward to corresponding advancements in treatment research that will in turn validate the correctness of these theories.

### 2.4. B Lymphocytes

Compared to T lymphocytes and NK cells, there is less research on the role and function of B lymphocytes in sepsis. By understanding the physiological functions of B lymphocytes, we have discovered that B lymphocytes have subpopulations similar to T cells, namely, effector B lymphocytes and Breg lymphocytes (regulatory B lymphocytes) [62]. Effector B lymphocytes are divided into two major subgroups, B1 and B2, and they play important roles in both innate and adaptive immunity [63]. (Figure 3) B1a cell-derived innate response activator B cells (IRA-Bs) and other cells-derived plasma cells (ASCs), along with memory cells, are the main terminally differentiated cells in the immune response. They, respectively, play roles in producing inflammatory factors (such as GM-CSF) and generating antibodies in the organism. In addition, B cells also have antigen-presenting functions and help activate CD4+ T cells [64].

In the collected clinical studies, there is a lack of meaningful results regarding B lymphocyte counts. However, interesting phenomena have been discovered through in-depth research on IRA-B cells and ASC cells. Animal experiments have shown that IRA-B cells exert a protective effect in sepsis mediated through the growth factor granulocyte-macrophage colony-stimulating factor (GM-CSF) [65]. In the human body, GM-CSF participates in innate immunity against infections through various pathways [66], including promoting phagocytosis by neutrophils, regulating the defensive factor IgM, and promoting the survival of IRA-B cells through autoregulation. Unfortunately, experiments delivering GM-CSF to the peripheral circulation to improve the prognosis of sepsis patients have not achieved the expected results. Further research is warranted and awaited. ASCs (antibody-secreting cells) are the foundation of humoral immunity [67]. Xu, Huihui, and colleagues [66] found a significant increase in ASCs in sepsis patients compared to healthy controls in clinical experiments. At the same time, they detected the increased proliferation of CD4+ T cells and the secretion of pro-inflammatory cytokines IFN-γ and IL-17 in sepsis patients. Xu Huihui and colleagues attributed this to the increased number of ASCs. Moreover, experiments observed the lowest point in the number of IgA+ ASCs on the third day. At this time point, the number of ASCs in the deceased group was significantly lower than that in the survival group. Finally, based on ROC survival curve analysis, ASCs were found to be the best model for predicting 28-day mortality in septic shock. The role of Breg cells in sepsis remains to be studied, and the coordination and regulation of different B lymphocyte subsets are still not fully understood. We look forward to breakthroughs in this direction in the future.

### 2.5. Dendritic Cells

Dendritic cells (DC) do not belong to the lymphocyte category in histological classification. Instead, they are part of the mononuclear phagocyte system along with monocytes and macrophages [68]. However, during the literature search, it was found that a significant proportion of research is devoted to DC, and multiple experimental results have confirmed the important role of DC in sepsis. This realization highlights the significance of this field and the need for further attention. Aspergillus fumigatus is an opportunistic pathogen. In the experiment conducted by Benjamim et al. [69], A. fumigatus was injected into mice with cecal ligation and puncture (CLP)-induced sepsis and mice that underwent sham surgery (abdominal incision only). The septic mice rapidly succumbed to fumigatus l infection, while the sham surgery mice exhibited tolerance to fumigatus infection. Furthermore, the septic mice were injected with bone marrow-derived dendritic cells (DC) by Benjamim et al., and the same experiments were repeated. The results showed that septic mice were protected from lethal fungal infection. The important role of antigen presentation in the pathogenesis of sepsis has been described [70]. According to the provided information [70], dendritic cells (DC) that display high levels of major histocompatibility complex class II (MHC II) molecules and co-stimulatory molecules are particularly effective at activating T cells. In comparison to other antigen-presenting cells (APCs), these DCs possess stronger antigen-presenting capabilities. The study conducted by Flohé et al. [71] involved extracting dendritic cells (DCs) from the spleens of mice that were subjected to cecal ligation and puncture (CLP)-induced sepsis, as well as mice that underwent sham surgery. The data showed that the number of DC in the spleens of septic mice was only 20% of that in the spleens of sham surgery mice. Furthermore, equal amounts of DC extracted from septic mice and sham surgery mice were separately co-cultured with inactivated T cells to observe T cell proliferation. The results showed significantly higher T cell proliferation in the sham surgery group compared to the septic mice, indicating that the antigen-presenting capability of DC in septic mice is impaired. According to the relevant articles, dendritic cells (DC) in the body mainly exist in an immature state and are responsible for antigen presentation [71]. The decrease in splenic DC during the process of sepsis is primarily caused by the apoptosis of immature DC [72]. 

The recent review by Bouras and colleagues [73] has described the role of DC cells in immunosuppression during sepsis. DC cells exist in an immature state during homeostasis, and upon encountering an antigen they differentiate into specialized mature DCs, losing the ability to recognize new antigens. Nonetheless, immature DCs are constantly being produced, making it possible to continuously recognize various antigens. However, in the case of sepsis, the newly generated immature DCs are subject to local modulation by the severe inflammation-induced immune panic, rendering them susceptible to tolerogenic signals. These tolerogenic DCs are unable to produce immunogenic functions in response to subsequent infection threats, and there is a reduction in bacterial clearance during episodes of secondary pneumonia. This forms a crucial part of the immunosuppression that occurs in the later stages of sepsis [74]. Bouras and colleagues further suggest that corticosteroids/steroids may inhibit the overactivation of DCs during severe inflammatory stages, as well as limit the formation of an immune panic microenvironment, with the purpose of containing immunoparalysis [53,55,75,76,77,78].

## 3. Conclusions

After extensive reading and contemplation, we believe that simply comparing absolute lymphocyte counts is meaningless. However, in addition to the absolute lymphocyte count, congenital lymphocyte subsets, effector T cells, B lymphocytes, and dendritic cells have shown certain research value in relevant studies. On the one hand, these specific subsets are not affected by differential components in the same way as absolute lymphocyte counts, and on the other hand, the manipulation of the specific subset numbers is more precise and feasible. Further in-depth research is necessary. Finally, we believe that determining the onset and progression stages of sepsis is of utmost importance. This will not only help further refine clinical research but also have significant implications for the advancement of clinical treatment. Although the current clinical application of lymphocyte counting and quality is very limited, as our research on lymphocytes and their subsets becomes more in-depth, I believe that in the near future, lymphocyte count and quality assessment can help clinicians make better clinical decisions, especially playing an important role in the field of sepsis.

## Figures and Tables

**Figure 1 ijms-25-01902-f001:**
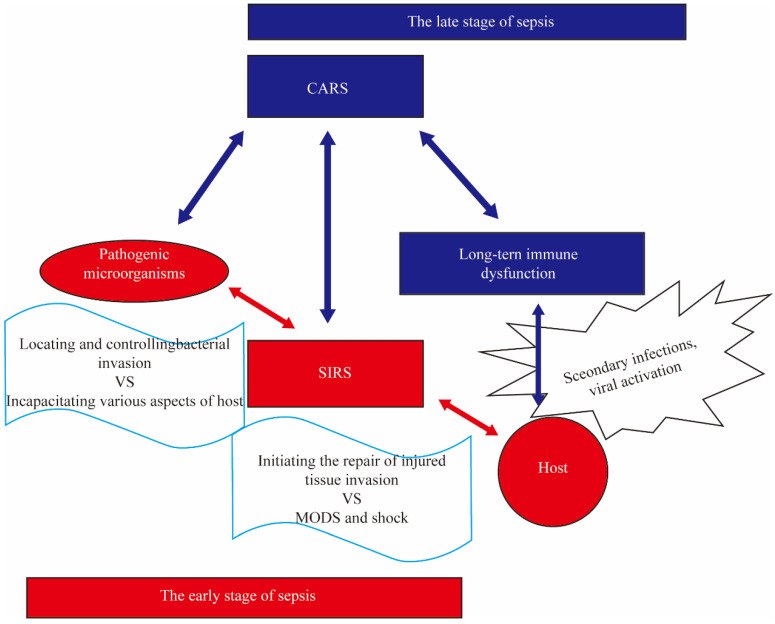
The Simplified Diagram of Compensatory Anti-Inflammatory Response Syndrome. The stimulation of pathogens and their products such as lipopolysaccharide (LPS) can induce systemic inflammatory response syndrome (SIRS) in the body, leading to multiple organ dysfunction syndrome (MODS), shock, and even death. The surviving patients during this stage will go through a compensatory anti-inflammatory response syndrome (CARS) stage, characterized by immunosuppression, which results in long-term immune dysfunction known as immune paralysis. Compared to patients without immune paralysis, those with immune paralysis are more prone to secondary infections, increased viral activation, and a decreased 5-year survival rate.

**Figure 2 ijms-25-01902-f002:**
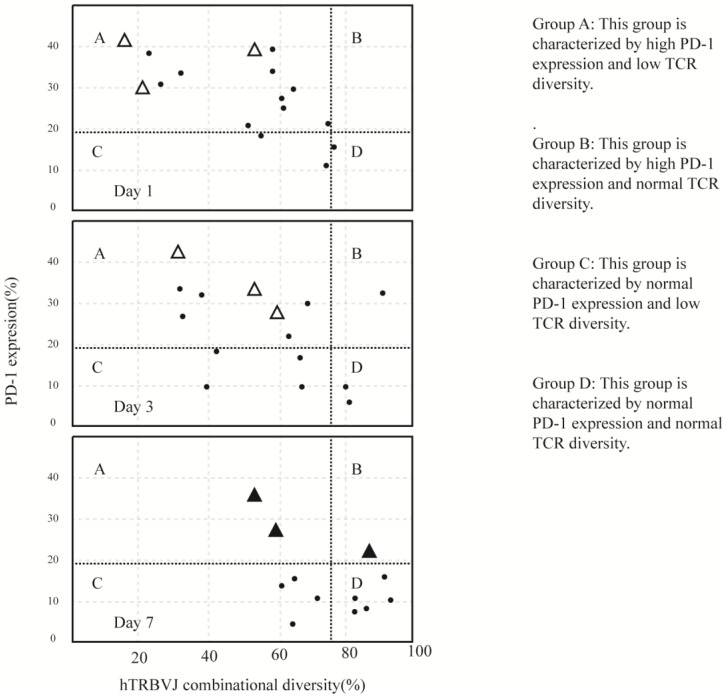
TCR diversity plotted versus PD-1 expression on quadrant charts for days 1, 3, and 7. (The data are from the experiment conducted by Atsutoshi Tomino et al. [56]). In each of the quadrant charts for days 1, 3, and 7, there are four zones defined by the normal thresholds of TCR diversity and PD-1 expression. These thresholds are determined based on the lower quantile data from healthy volunteers for TCR diversity (78.3%) and the upper quantile data for PD-1 expression (19.5%). The patients with CMV reactivation who unfortunately passed away are indicated by closed triangles on day 7 and by open triangles on days 1 and 3. The individual values of TCR diversity and PD-1 expression for each patient are presented in the charts.

**Figure 3 ijms-25-01902-f003:**
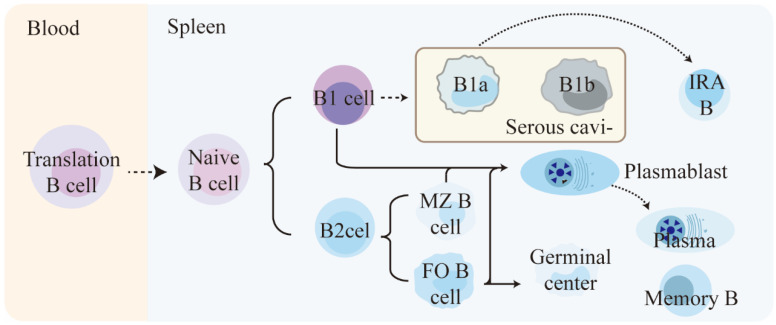
Differentiation process of mouse B cell subpopulations under physiological conditions. (The content shown in the figure is referenced from the article by Sandra Romero-Ramírez [63]). This passage describes the maturation and differentiation of B cells in the spleen. Naive B cells first mature from immature/transitional stages in the spleen and give rise to two lineages, B1 cells and B2 cells. B1 cells differentiate into B1a and B1b cells, while B2 cells in the spleen form FO B and MZ B cells. FO B cells are found in the follicles of lymphoid organs and MZ B cells reside in the splenic marginal zone. When mature B cells encounter antigens, they may differentiate into short-lived plasmablasts or enter the germinal center reaction. Eventually, they differentiate into long-lived plasma cells or memory B cells. Additionally, serosal B1a cells can migrate to the spleen and become IRA B cells, which are characterized by GM-CSF secretion.

**Table 1 ijms-25-01902-t001:** Article summary related to absolute lymphocyte count in sepsis.

Reference Numbering	Publication Date	Number of Included Patients	Do Lymphocytes Decrease (+: yes; -: no)	Criteria for Decreased Lymphocytes	Is a Decrease in Lymphocytes Related to Prognosis? (+: yes; -: no)
[15]	2000	40	+	None	-
[16]	2011	40	+	None	-
[17]	2014	335	+	On the fourth day after diagnosis of sepsis, the absolute lymphocytes counts are ≤0.6 cells/uL*10^3^.	+
[18]	2019	124	+	<1500 cells/uL	+
[19]	2019	100	+	<1.0*10^9^/L	+
[20]	2021	216	+	None	+
[21]	2021	2203	+	<724 cells/uL	+
[22]	2023	243	+	None	+

## Data Availability

Not applicable.

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
