# Peer review of "The Changes in the Quantity of Lymphocyte Subpopulations during the Process of Sepsis"

_ijms, 2024, doi:10.3390/ijms25031902_

Round 1

Reviewer 1 Report (New Reviewer)

Comments and Suggestions for Authors

This review aimed to ascertain the relationship between absolute lymphocyte count, innate lymphocyte subsets, effector T lymphocytes, B lymphocytes, dendritic cells, and the onset and prognosis of sepsis. While appreciating the topic for this review, following comments may please be considered;

·         The title needs to be revised, please see my suggestion in the manuscript comments.

·         The abstract is too short and needs to be re-written that should cover the highlights of this review.

·         Line 26: reference is too old (2008) please try to give recent scenario about sepsis as a leading cause of deaths in ICU

·         Line 37: please give reference

·         Line 52-53: An article of 2001 cannot be considered as a recent study. Please provide an updated reference.

·         Line 64-87: Please omit as this does not fit here and does not go with the flow of the review.

·         Line 89: Please provide details of the review process. PRISMA etc.

·         The article should not have results and discussion headings, rather subheadings to discuss in detail about the lymphocyte subpopulation and finally the conclusion section.

·         Line 115: please re-write ‘mortality of sepsis’

·         In table 1, why is there a huge gap in review of literature between 1993 and 2011 by the authors?

·         Line 152: Give some recent advancements in research on these lines

·         Line 174: TGF (Transforming growth factor) please specify first before mentioning the abbreviation

·         Line 228: Omit ‘Where there is commonality, there will be differences’

·         Line 259-268: can be part of figure legend

Discussion can be omitted

Author Response

Thank you very much for the comments on my article. I have made the relevant modifications, please take a look at the revised manuscript that I've re-uploaded. If there are any further issues, please do not hesitate to point them out.

Reviewer 2 Report (New Reviewer)

Comments and Suggestions for Authors

This is a very concise and well written review on a complex topic.  The authors focus on studies demonstrating the impact of  lymphocytopenia  and different lymphocyte  subset changes during sepsis .  

Rec and minor  corrections to table 

Table study Reference 14 The population is a mixed population no just septic patients 

In the table the authors should describe septic subgroup in this study and whether these pts had lower lymphocyte counts and whether this was predictive of outcome.  

Study reference 21 these were all septic pts.  lymphopenic pts had higher incidence of septic shock change word sepsis to septic shock 

Suggestions 

Consider adding the immunosuppresive effects of  immature dendritic cells when induced by the inflammatory scare response and the role of corticosteroids inhibiting response and potentially preventing secondary infection.

"In mice, the restoration of the immunogenic functions of DCs restores the mucosal immune response to pathogens. In humans, the modulation of inflammation by glucocorticoid during sepsis or trauma preserves DC immunogenic functions and is associated with resistance to secondary pneumonia. Finally, we propose that the alterations of DCs during and after inflammation can be used as biomarkers of susceptibility to secondary pneumonia and are promising therapeutic targets to enhance outcomes of patients with secondary pneumonia"

see Bouras et al. Frontiers in immunology  doi: 10.3389/fimmu.2018.02590. eCollection 2018

A table reviewing the role of different lymphocyte subsets  the growth factors and cytokines that influence these changes during sepsis may be helpful 

I agree that following absolute lymphocyte counts although have prognostic implications are quite useless from a therapeutic standpoint.  However changes in lymphocytes subsets are.  Consider adding comments regarding endotyping  pts with sepsis according to changes in lymphocyte subsets based either by transcriptomic analysis  with  high throughput analysis such as are there populations of septic patients that can be classified as a inflammatory phenotype vs an adaptive immunosuppressed phenotype 

Author Response

Thank you very much for the comments on my article. I have made the relevant modifications, please take a look at the revised manuscript that I've re-uploaded. If there are any further issues, please do not hesitate to point them out.

Reviewer 3 Report (New Reviewer)

Comments and Suggestions for Authors

Dear Authors,

Thank you for your manuscript. Please let me highlight some points that need to be addressed:

Line 29.

As a result, global incidence and mortality rates of sepsis have shown a downward trend.

The decrease in incidence is contradicted (https://jtd.amegroups.org/article/view/34844/html) while I agree that mortality shows some decrease.

Figure 1. Please aling text with boxes! It looks strange to have letters outside the box.

Line 80-84

Please elaborate how immune tolerance is achieved, i.e. citokines, other inflammatory mediators.

Section for Methods is missing. Please describe how the search was designed, search phrases, databases, etc.

Line 146.

I think Cell should be written with a small c

Line 151

Either APC or antigen presenting cells – the brackets and abbreviation are unnecessary

Line 181

higher overall expression of Foxp-3” is a different font type, please correct!

Line 395.

I strongly disagree with the statement „In my opinion, the count of effector T lymphocytes is considered the most promising biomarker for sepsis.”. On the one hand it is a personal opinion of the author  on the other hand a cell count is definitely not a biomarker by definition.

Please provide some proof that you have permission to use Figure 2. and 3.

As this paper is a nice summary on lymphocyte involvement in sepsis, the authors themselves admit that this is not the method of choice in prediction of sepsis survival or severity. Could you please look into the prospect of using lmyphocyte count and quality in personal medicine, especially in septic patients?

Thank you!

Author Response

Thank you very much for the comments on my article. I have made the relevant modifications, please take a look at the revised manuscript that I've re-uploaded. If there are any further issues, please do not hesitate to point them out.

Round 2

Reviewer 1 Report (New Reviewer)

Comments and Suggestions for Authors

Authors have made satisfactory progress in the revisions as per the suggestions.

This manuscript is a resubmission of an earlier submission. The following is a list of the peer review reports and author responses from that submission.

Round 1

Reviewer 1 Report

Comments and Suggestions for Authors

The following review paper summarizes biomarkers based on white blood cells for the early detection and diagnosis of sepsis. It is well-organized for easy understanding and provides valuable information to its readers.

The overall structure, flow, and focus of the content in this review paper are good. However, a few minor revisions are needed, particularly with the figures.

1. In Table 1, It would be better to include references for each article with the new column.  

2. In Figure 2, displaying references for each group in the figure caption would be beneficial.  

3. Similarly, in Figure 3, references from the cited papers should be indicated in the figure caption (preferably with the main author's name and published year with journal name)  

4. In Figure 4, it would be better to display references for each of the three clinical outcome groups next to the graph.

Author Response

I have made the necessary revisions to the issues marked in the manuscript. Thank you for your patient guidance, and I will take this experience into account and pay attention to these issues in my future writing.

Reviewer 2 Report

Comments and Suggestions for Authors

1.     In discussion part lines 309-313, the authors detailed that “absolute lymphocyte count cannot be considered an ideal biomarker. First, there is a lack of standardized definition for lymphocyte reduction. The variation in defining criteria can lead to significant differences among enrolled patients, making it difficult to obtain convincing results even if the outcomes may be similar”. However, in conclusion (lines 368-369), the authors believe that using lymphocytes as biomarkers for the diagnosis and prognosis of sepsis is feasible! It would be worth clarifying this discrepancy for a better take home message.

2.     A defining pathophysiologic feature of sepsis is profound apoptosis-induced death and depletion of CD4+ and CD8+ T cells and PMID 29515037 have shown that the anti-apoptotic and lymphocyte function-enhancing cytokine IL-7 was well tolerated and reversed sepsis-induced lymphopenia in patients suggesting that this approach be further explored. On these lines, why the authors feel that absolute lymphocyte count cannot be considered an ideal biomarker?

3.     Figure 1 tilted “The antagonistic effect between pathogenic and host microorganisms” sounded bit irrelevant in the context of this review.

4.     This review was written in a manuscript format. Talking about Results, discussion sounded bit out of format!

5.     Several immune cells play crucial roles in pathophysiology of sepsis. As the authors prepared to emphasize on the “changes in the Quantity of Lymphocyte Subpopulations during the Sepsis” it would be worth being more specific and more detailed within that specific lines instead of taking about non-lymphocyte populations unless the authors change the title of the review

Comments on the Quality of English Language

This review  may fare better with better english

Author Response

(The authors gave the same response as above.)

Reviewer 3 Report

Comments and Suggestions for Authors

Yang et al. present a review article on "The Changes in the Quantity of Lymphocyte Subpopulations 2 during the Process of Sepsis". It is overall well written and the topic is of interest. I think the manuscript can still be improved. It focuses on the quantitative changes in lymphocyte subsets and DCs during sepsis, but also touches on pathogenetic mechanistics of leukocytes during sepsis.
To make this more straightforward, I would suggest to either focus more clearly on the leukocyte numbers as a potential diagnostic tool, which I would suggest. In that case, the more mechanistic considerations on some of the subsets could be removed.
Or, if the pathogenetic considerations are to be kept, please add more in-depth and up-to-date explanations on immunological mechanisms of sepsis as a highly dynamically regulated process.
In the discussion the authors correctly state, that lymphocyte counts are not specific. Later they propose that they see potential as a diagnostic tool, they should elaborate more why (e.g. the potential of specific subsets as opposed to global numbers...)
Last, if DCs are mentioned, maybe change the title stating that it considers adaptive immunity... Or if the authors decide to focus more on cell numbers as a diagnostic tool, why not add a section on monocytes and granulocytes...

In addition to these general remarks, I have some specific comments:
Figure 2: Where is this data from? Own data by the authors?
Figure 3: Serous cavity?
Figure 4: What does "days post injust" mean? What is the basis for these numbers? Own data? Then the reader needs details on the patient cohort. Or from the literature? Then it should be cited. This must be explained in the figure legend. Or is it just an exemplary drawing?
Line 102: "Congenital lymphocyte subsets": It is unclear what that is. Also mentioned in the conclusion.
Line 148: "higher concentrations of Foxp-3": Rather: Higher overall expression of Foxp-3.
    Also: This experiment is described in much more detail than the others, I would shorten the description (Treg were isolated from patients...)
Line 150: "The above experimental results support the negative regulatory role of Treg cells in sepsis". I have several issues with this sentence. What is a negative regulatory role?
    They are supposed to play a regulatory role. If this wants to state that they have a negative impact on survival, I must disagree, as this is a mere correlation in a highly dynamic process, as stated correctly below, Treg activity can be beneficial or harmful depending on the stage of disease.
Line 210: "Therefore, it supports the coexistence of pro-inflammatory and anti-inflammatory responses during sepsis". I think it is commonly accepted, that both pro- and antiinflammatory mechanisms are part of the complex regulation during sepsis
Line 286: "The septic mice rapidly succumbed to viral infection, while the sham surgery mice exhibited tolerance to viral infection". Isn't that study about Aspergillus? And not viral infection? Also, I understood that DCs are given intratracheally, please check the reference.
Line 315: What is "Article 1"?
Line 331: And viral infections

Comments on the Quality of English Language

Minor revisions would improve the language. Some terms are unusual in the field (e.g. congenital lymphocytes)

Author Response

(The authors gave the same response as above.)

Reviewer 4 Report

Comments and Suggestions for Authors

Dear Authors,

I would like to congratulate you with the submitted manuscript.

I find the subject interesting but requires profound analysis as presented analysis in not expanding our horizons on the topic.

In your review you focused on sepsis but not presented the possible relation between the causative agents of sepsis and the lymphocytic response.

I would suggest revising the manuscript and present the lymphocyte response related to causative factors.

Kinds

Author Response

(The authors gave the same response as above.)

Round 2

Reviewer 2 Report

Comments and Suggestions for Authors

I see no revision as such in response to the comments or recommendations made. Moreover, authors response stating that "I will take this experience into account and pay attention to these issues in my future writing" may not sound okay with the current review article. I hope the authors agree with my opinion. 

Comments on the Quality of English Language

This review article will certainly benefit with substantial editing.

Reviewer 4 Report

Comments and Suggestions for Authors

Dear Authors,

thank you for NOT correcting the manuscript according to my suggestions.

kinds

R